# Formulation of tunable size PLGA-PEG nanoparticles for drug delivery using microfluidic technology

**Adrianna Glinkowska Mares**[1], **Gaia Pacassoni**[1,2], **Josep Samitier Marti**[1,3,4], **Silvia Pujals**[1,3], **Lorenzo Albertazzi**[1,5] *

**1** Institute for Bioengineering of Catalonia (IBEC), The Barcelona Institute of Science and Technology (BIST), Barcelona, Spain, **2** Department of Mechanical and Aerospace Engineering, Politecnico di Torino, Torino, Italy, **3** Department of Electronic and Biomedical Engineering, Faculty of Physics, University of Barcelona, Barcelona, Spain, **4** Networking Biomedical Research Center in Bioengineering, Biomaterials and Nanomedicine (CIBER-BBN), Madrid, Spain, **5** Department of Biomedical Engineering, Institute for Complex Molecular Systems (ICMS), Eindhoven University of Technology, Eindhoven, The Netherlands

* l.albertazzi@tue.nl

**Data Availability Statement:** The data used in this study can be accessed at https://data.4tu.nl/articles/dataset/Data_underlying_the_publication_Formulation_of_tunable_size_PLGA-PEG_

## Abstract

Amphiphilic block co-polymer nanoparticles are interesting candidates for drug delivery as a result of their unique properties such as the size, modularity, biocompatibility and drug loading capacity. They can be rapidly formulated in a nanoprecipitation process based on self-assembly, resulting in kinetically locked nanostructures. The control over this step allows us to obtain nanoparticles with tailor-made properties without modification of the co-polymer building blocks. Furthermore, a reproducible and controlled formulation supports better predictability of a batch effectiveness in preclinical tests. Herein, we compared the formulation of PLGA-PEG nanoparticles using the typical manual bulk mixing and a microfluidic chip-assisted nanoprecipitation. The particle size tunability and controllability in a hydrodynamic flow focusing device was demonstrated to be greater than in the manual dropwise addition method. We also analyzed particle size and encapsulation of fluorescent compounds, using the common bulk analysis and advanced microscopy techniques: Transmission Electron Microscopy and Total Internal Reflection Microscopy, to reveal the heterogeneities occurred in the formulated nanoparticles. Finally, we performed in vitro evaluation of obtained NPs using MCF-7 cell line. Our results show how the microfluidic formulation improves the fine control over the resulting nanoparticles, without compromising any appealing property of PLGA nanoparticle. The combination of microfluidic formulation with advanced analysis methods, looking at the single particle level, can improve the understanding of the NP properties, heterogeneities and performance.

## Introduction

Since more than three decades polymeric nanoparticles (NPs) are investigated as drug delivery systems (DDSs) for treatment of several diseases with a focus on cancer [1–6]. They carry a

nanoparticles_for_drug_delivery_using_
microfluidic_technology/14572362.

**Funding:** LA and SP thank the Spanish Ministry of Science and Innovation (PID2019-109450RB-I00/AEI /10.13039/501100011033) and the Generalitat de Catalunya through the CERCA program and 2017 SGR 01536. The authors also acknowledge the the foundation Obra Social La Caixa (ID 100010434) and the European Research Council (ERC- StG-757397). JSM has support from the CERCA Programme and by the Commission for Universities and Research of the Department of Innovation, Universities, and Enterprise of the Generalitat de Catalunya (2017 SGR 1079). This work was partially funded by the Spanish Ministry of Economy and Competitiveness (MINECO) through the projects MINDS (Proyectos I+D Excelencia + FEDER): TEC2015-70104-P, BIOBOT (Programa Explora Ciencia / Tecnología): TEC2015- 72718-EXP and EuUONANOMED II PCIN-2016-025. The project that gave rise to these results received the support of a fellowship from "la Caixa" Foundation (ID 1000010434). The fellowship code of AGM is LCF/BQ/DI17/11620054. This project has received funding from the European Union's Horizon 2020 research and innovation programme under the Marie Skłodowska-Curie grant agreement No. 713673.

**Competing interests:** The authors have declared that no competing interests exist.

**Abbreviations:** ACN, Acetonitrile; AS, antisolvent; CFD, Computational Fluid Dynamics; DDS, drug delivery system; DLS, Dynamic Light Scattering; DMEM, Dulbecco's Modified Eagle Medium; DOX, Doxorubicin; EE, Encapsulation Efficiency; EMA, European Medicines Agency; EPR, Endothelial Permeability and Retention; FDA, Food and Drug Administration; HFF, Hydrodynamic Flow Focusing; NPs, Nanoparticles; NR, Nile Red; PdI, Polydispersity Index; PDMS, Poly (dimethylsiloxane); PEG, Polyethylene Glycol; PLGA, Poly(lactic-co-glycolic) acid; S, solvent; TEM, Transmission Electron Microscopy; TIR, Total Internal Reflection.

promise of improved therapeutic effect in the view of their unique properties, such as size, shape, porosity, charge or modifiable surface [7–10]. One of the key features of a nanocarrier is the size that promotes escape through the gaps in tumorous vasculature, that are the fundament of passive drug delivery design [11, 12]. Furthermore, the NP surface can be modified to limit the adsorption of proteins and provide longer circulation [13, 14]. Importantly, polymeric nanocarriers can encapsulate various therapeutic molecules, to protect them from premature deactivation or to avoid undesired cytotoxicity [15, 16]. Another advantage of drug encapsulation is the improved bioavailability of poorly water-soluble compounds, which are the majority of active pharmaceutical ingredients [17, 18]. Encapsulated payload can be released in a controlled way upon the degradation of the nanocarrier's polymeric matrix, offering sustainable delivery of the drug, what is important for a positive therapeutic effect [19, 20]. Additionally, the NPs can actively target specific sites as a result of the surface functionalization with targeting ligands [21–23]. The combination of these features creates a promising drug delivery system with improved pharmacokinetics and better therapeutic outcome comparing to the currently available solutions [24].

One of the most studied polymers in drug delivery is Poly(lactic-co-glycolic) acid (PLGA), which has already been FDA- and EMA-approved for various applications [25, 26]. PLGA breaks down to the lactic and glycolic acids, ensuring the biocompatibility and biodegradability required for safe use in humans [27, 28]. The tunability of the lactic to glycolic acid ratio and the polymer chain length, allow to adjust the polymer matrix degradation providing controlled drug release, ranging from days to years [27, 29]. The summary of PLGA properties makes it the most versatile polymer in parenterally administered drug delivery systems, currently used in the form of emulsions, microparticles and implants [30–32].

In the course of PLGA-based achievements in the field of drug delivery, it naturally became engaged in the development of a nanoparticle based systems. Block copolymers, such as PLGA with polyethylene glycol (PEG) are widely used to formulate NPs. The conjugated PEG allows to minimize the use of formulation stabilizers and importantly, it decreases the formation of protein corona after administration and results in extended systemic circulation of the NPs [33–36]. Furthermore, the PEG chain terminated with targeting ligands, demonstrated positive results in *in vitro* studies [22, 37].

The PLGA-PEG amphiphilic diblock copolymer can be folded into NPs via self-assembly, by addition of the polymer solution into a miscible antisolvent (AS). In this process the PLGA-PEG solubility suddenly decreases and it starts to precipitate, forming kinetically frozen NPs [38–40]. Therefore in the nanoprecipitation method, the uniform mixing of both liquids is crucial to obtain homogenous NPs [41–43]. Ideally, the process should be robust and reproducible to control the NP properties, which later dictate their fate in human body and determine success of the DDS.

In the last two decades microfluidic technology appeared as an interesting alternative in the formulation of NPs. In contrary to the manual bulk mixing method, it offers precise and homogenous mixing of the polymer solution with the antisolvent phase [44–48] This approach uses microfluidic chips similar to these known for microparticle formation [49–52]. It is based on the laminar flow of the liquids confined in the microchannels and results in hydrodynamic flow focusing (HFF) of the solvent (S) phase. The restricted volume allows to limit the diffusion distance of solvent and antisolvent, therefore it regulates the mixing of the two liquids that can be further controlled by modification of stream flow rates or chip design [53–56]. The fine control over this process results in tunability of the NP properties, such as its size, surface characteristics or crystallinity [57–59]. It reduces batch-to-batch variability and enables the investigation on how the NPs properties can be tailored upon formulation parameters, what was not possible within the bulk mixing method [45, 60, 61]. Another advantage in the use of

microfluidic chips is the possibility of simulations of fluid dynamics by computational methods. The diffusion can be visualized to better understand process parameters and to improve the experimental planning [62, 63].

Taking into account the successfulness of PLGA in DDS, the rapidly developing microfluidic technology was quickly introduced into the formulation of PLGA micor- and nanoparticles. Different chip geometries, formulation compounds and PLGA conjugates were explored, demonstrating improved control over the particle properties and resulting in various sophisticated PLGA-based nanocarrier systems [64–66].

In our work we investigated the impact of the microfluidic chip assisted HFF nanoprecipitation on the particle size tunability and cargo loading in comparison to the manual bulk mixing method. We performed surfactant-/stabilizer-free formulation of PLGA-PEG NPs in a microfluidic chip at different S and AS flow rates, and by a manual dropwise addition of the S to the AS phase at parallel volumes. We studied encapsulation efficiency of fluorescent molecules with different hydrophobicity. Obtained NPs were characterized using bulk and single-particle methods for general and more detailed information. Our data confirmed that the particle diameter can be tailored and controlled with the use of microfluidic chip in the formulation process. The advanced microscopy techniques used to characterize single NPs allowed us to reveal heterogeneities present in the batches and indicated more homogenous dye loading in the HFF-formulated NPs. Furthermore, the bioevaluation of NPs obtained via microfluidic formulation demonstrated cell internalization and biocompatibility.

## Results and discussion

### Experimental setup of microfluidic hydrodynamic flow focusing and manual bulk formulation

Particles were formulated via nanoprecipitation method, using manual dropwise addition in bulk mixing and a microfluidic setup facilitating hydrodynamic flow focusing (Fig 1a and 1b, respectively). The polymer was dissolved in acetonitrile (ACN), an organic solvent miscible with water, which was the antisolvent for PLGA-PEG [67]. The diffusion of ACN into water reduces solubility of the polymer and, as a response to this change of environment, the amphiphilic blocks rapidly self-assemble into NPs. In this process the PLGA blocks concentrate in the core of the NP and most of the PEG chains are exposed on the surface, forming the corona [40], as schematically illustrated in the Fig 1b. The nanoparticle formation is divided into three phases: nucleation, growth, and aggregation. The pivotal role in the output of nanoprecipitation plays the mixing, responsible for homogenous supersaturation inducing polymer nucleation. Poorer mixing results in low nucleation rate and a growth of larger particles because the polymer aggregation occurs in the presence of higher fractions of organic solvent ($\tau_{mix} > \tau_{agg}$) and, if mixing occurs faster than the time scale for aggregation ($\tau_{mix} < \tau_{agg}$), the aggregation phase takes place when mixing is almost complete and more nuclei are formed [68–70].

Microfluidic chips are tools, that can aid spatial and temporal separation of the above-mentioned three phases of particle formation [68]. In our setup the main part is the chip with three inlets and one outlet (microchannels of 200 μm width and 60 μm height) connected to a syringe pump with a capillary tubing. We injected the solvent phase into the central inlet and controlled its stream width (at the outlet) by the flow rate of laterally injected antisolvent, as illustrated in the Fig 1b. Changes in the AS flow rate resulted in a range of solvent to antisolvent flow ratios, and the faster the lateral flow rate, the narrower the central stream and smaller the diffusion distance. To visually demonstrate it, we employed computational fluid dynamics (CFD) using parameters corresponding to water and ACN for the AS and S phase. The Fig 1c illustrates the impact of changing the AS flow rate in the microchip inlets on the solvent

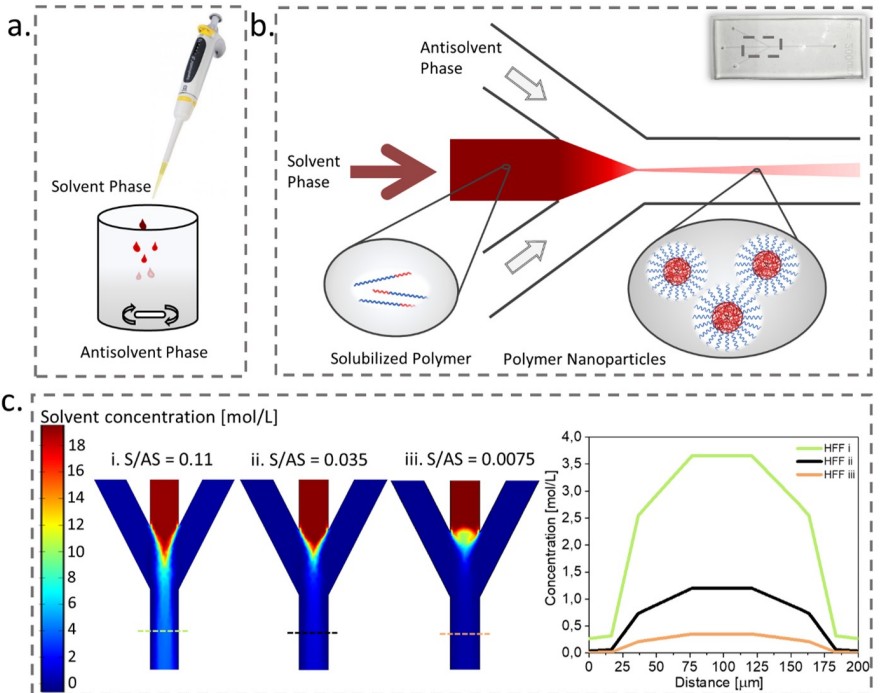

**Fig 1. Formulation of nanoparticles via nanoprecipitation.** a. manual bulk mixing, b. hydrodynamic flow focusing, c. fluid dynamics simulation for three different solvent to antisolvent flow rates (S/AS)and the solvent concentration (i-iii) at the outlet cross-section 0,3 mm below the junction (marked by the dashed line in the simulations on the left).

concentration gradient in the outlet channel. With the decrease of S/AS, we can observe significant decrease of the solvent concentration right after the junction of the inlet channels. As a result of narrowing the solvent stream, with the increase of AS flow rate, the solvent diffusion can occur more rapidly and promote faster mixing of the two miscible liquids.

## Particle diameter tunability using microfluidic chip

The particle size resulting in the nanoprecipitation process is dictated by the mixing of S and AS phase. Faster mixing of the two phases leads to a locally lower fractions of organic solvent, what yields smaller NPs, kinetically locked in the non-solvent environment [69, 70]. To probe the formulation of size-tailored NPs, resulting from the controlled diffusion of S into the AS phase, we used the above-described chip and manual droplet addition in a bulk mixing for the control. We formulated the NPs with the microfluidic device using the constant flow rate of 5 μL/min for the polymer solution and an adjustable lateral inflow of the AS, ranging from 20 to 330 μL/min, and resulting in the S/AS ratios between 0.0075 to 0.11. Within these parameters we calculated the mixing time $\tau_{mix}$ in our system, which should be lesser than the aggregation time $\tau_{agg}$ to control the particle size. We found values from 0.25 ms to 44 ms for the extremes of the tested S/AS range (see S1 File), which are lower than estimated unimer penetration time leading to aggregation [71]. To compare the HFF results with the bulk nanoprecipitation method, we formulated the NPs by manual addition of the solvent phase into the antisolvent, controlling the mixed volumes to achieve comparable S and AS final volumes in both approaches. Next, we measured the NPs diameter using dynamic light scattering (DLS) and imaged them with negative-staining transmission electron microscopy (TEM). The results of the DLS measurement shown in the Fig 2a

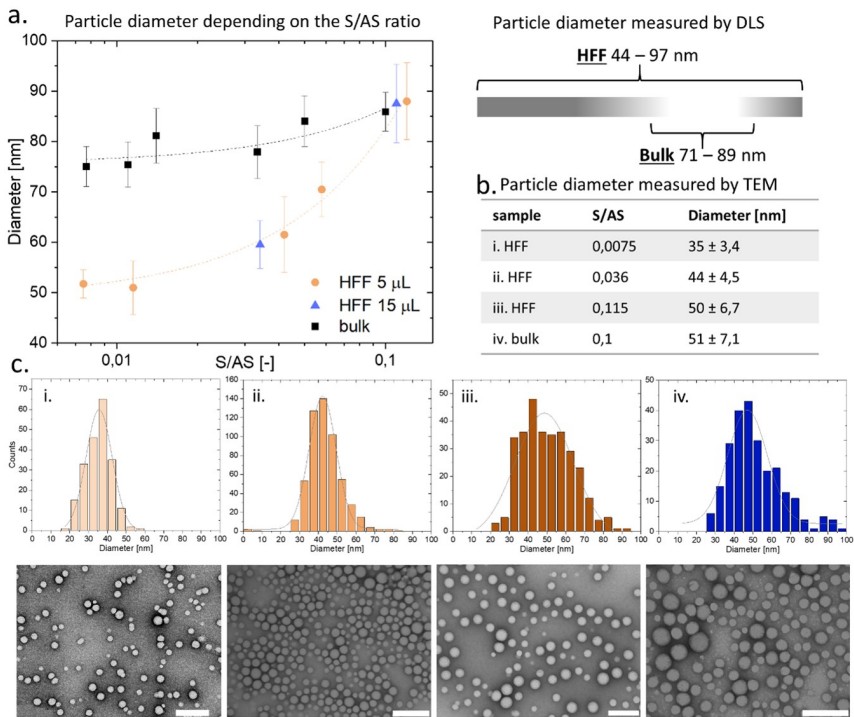

**Fig 2. Particle size analysis.** a. Particle diameter measured by DLS, (left) represented in the function of the S/AS flow ratio and (right) schematically illustrated dependency on the formulation method (bulk and HFF—hydrodynamic flow focusing); b. table with an average particle diameter obtained from analysis of TEM images; c. Particle size distribution obtained from TEM analysis (top row) and the corresponding TEM images (bottom row), scale bar 200 nm.

demonstrated how the ratio of the S and AS phase flow dictates the particle hydrodynamic diameter. The lower the S/AS flow rate, the smaller the NPs size, ranging here from 44 to 97 nm. In the manual dropwise addition process we obtained particles in size of 71 to 89 nm, with only a slight diameter change in response to the S/AS volume modifications. The HFF formulation allowed to obtain even 50% smaller NPs comparing to the dropwise addition process and importantly, it permitted to tune the size of the NPs in the response to the S/AS flow ratio, what corresponds to the particle size tunability in response to the flow parameters reported in the literature [51, 60, 65]. We performed similar HFF experiment with higher absolute solvent flow rate ($S_{flow\ rate}$ = 15 μL/min) to check if the size tailoring trend is maintained at different flow rate values. However, the increase of corresponding AS flow rates (to maintain the same S/AS ratio as tested with the $S_{flow\ rate}$ = 5 μL/min) was not suitable for our chip, due to its integrity loss resulting from too high total flow rates. Nevertheless, the hydrodynamic diameter of NPs formulated with S flow rate of 15 μL/min at a narrower range followed the same pattern as the NPs diameter formulated with S flow rate of 5 μL/min, as it is shown in Fig 2a. All the performed formulations yielded monodisperse NPs with the polydispersity index (PdI) in the range of 0,06–0,15. Overall, we used two methods of nanoprecipitation-based particle formulation and we were able to achieve a broader range of NPs sizes using the microfluidic chip. This approach demonstrated how the particle size can be tuned (44–97 nm for HFF comparing to 71–89 nm for manual formulation) without modifying the copolymer molecular weight or other chemical or physical properties (such as Zeta Potential, S1 Fig). It is especially important to control the particle size, as their biodistribution and cellular uptake depends on this feature [35, 72].

DLS measurement rapidly provides the information about an average hydrodynamic diameter and dispersity of a batch, however it does not reveal details of a single NP. To look at the individual particle's size and its distribution we used transmission electron microscopy (TEM). The particles were deposited on carbon grids with uranyl acetate negative staining, dried and imaged. We measured the diameter of a minimum 200 particles for each batch and compared the results of formulations with different S/AS flow ratios, as well as between the two methods used for nanoprecipitation as demonstrated in Fig 2b and 2c. The analysis on the single particle level demonstrated the same trend as the hydrodynamic size measurement with the DLS, confirming the size tunability achieved with the microfluidic device, with the average particle diameter of 35 nm, 44 nm and 50 nm for the S/AS flow ratio of: 0,0075; 0,036; and 0,115, respectively. We found the NP average diameter values smaller for TEM image-based measurement than for the DLS method, which can be explained by the technique differences (DLS measures hydrodynamic diameter in suspension, meanwhile in TEM images we measured the gyration radius of dried particles). Nevertheless, the TEM analysis were coherent with previously observed DLS measurement-based trend. The average diameter of NPs formulated using the microfluidic device was dependent on the S/AS flow ratio, yielding smaller size at lower S/AS. Additionally, the TEM images helped us to visualize the heterogeneity among the particles, in the Fig 2c (bottom row), we could identify particles in a range of 15–60 nm (in batch i), meanwhile the average size was 35 nm, however the dispersity of each batch was on the level of ~10%, similar to the values indicated by the DLS (PdI = 0,06–0,15). We also observed the particle morphology to be heterogenous, some of the NPs displayed distinctive core-corona structure, resulting from the separation of the PLGA and PEG blocks in the self-assembly process (S2 Fig). The differences in these features, revealed only when analyzing single particles, contribute to the overall performance of the nanocarriers in *in vitro* or *in vivo* tests. However, it is important to understand if the outcome often taken as the "average" information indeed is represented by most of the NPs in the bulk [73].

## Encapsulation of fluorescent compounds

Upon the self-assembly of the amphiphilic block co-polymers into the NPs, the PEG chains become exposed in the surface of the nanoparticle, meanwhile the PLGA blocks are folded in the core. A spontaneous encapsulation of hydrophobic molecules present in the solvent phase occurs during the particle formation [18]. This is a common strategy to encapsulate drugs into PLGA formulations [74, 75]. We performed a series of formulations with different fluorescent compounds added to the solvent phase, aiming to investigate the encapsulation efficiency (EE) for both: manual and HFF formulation method. We loaded the nanocarriers with 1,1'-Diocta-decyl-3,3,3',3'-Tetramethylindocarbocyanine perchlorate (DiI), Nile Red (NR) and Doxorubi-cin (DOX), of which all have fluorescent properties, however different hydrophobicity from the most to the least hydrophobic listed, respectively. We formulated the particles by the manual bulk and the HFF nanoprecipitation using two S/AS to obtain different diameter NPs, then we collected them, washed and measured the EE. The absorbance measurement for each loaded compound revealed, that the highest EE of ~80% is associated with the most hydrophobic molecule (DiI), and lowest value of ~15% was measured for DOX. Similar values were found regardless the formulation method or the particle diameter (Fig 3a). The spontaneous entrapment depended here on the solubility of the compound in the antisolvent phase, therefore we measured the decreasing process yield for Nile Red and DOX. For therapeutic application it would be necessary to improve the DOX loading into the NPs, however in this study we investigated the trends regarding the encapsulation of different fluorescent molecules within the proposed nanoprecipitation protocol. In the literature there are strategies demonstrating

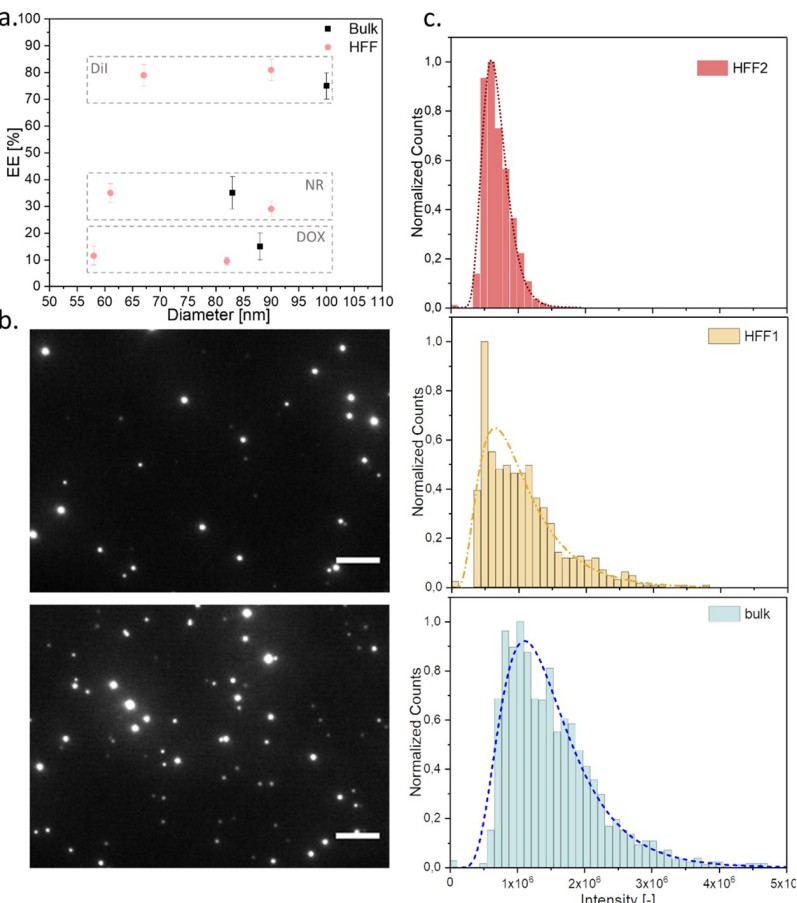

**Fig 3. Encapsulation of fluorescent compounds.** a. EE for DiI, NR and DOX loaded NPs, formulated with HFF and bulk nanoprecipitation, b. TIRF images of DiI loaded NPs formulated with HFF and bulk method (top: HFF1 = 95 nm, bottom: bulk = 89 nm), scale bar 5 μm. c. Analysis of emitted fluorescence intensity per particle for DiI loaded NPs, formulated with HFF and bulk methods, size by DLS: HFF1 = 95 nm, HFF2 = 71 nm, Bulk = 89 nm.

improved DOX loading into PLGA-based NPs, including use of emulsification or modified nanoprecipitation formulation method, hydrophobization of DOX or its conjugation to the polymer chains [76–78].

Parallel to the previous size measurements, the EE bulk measurement was followed by an analysis on a single particle level, using total internal reflection fluorescence (TIRF) microscopy. The DiI loaded NPs, formulated by HFF and bulk nanoprecipitation, were imaged using TIRF microscopy taking into account the dye's good EE. The high signal to noise ratio allowed us to visualize individual NPs and quantify their fluorescence, proportional to the molecule encapsulation. The acquired images revealed heterogenous fluorescence intensity emitted by the particles in the same field of view as can be seen in Fig 3b (top image—HFF, bottom—bulk formulation). To understand the difference, we imaged at least 800 NPs per formulation method, and quantified the emitted fluorescence intensity per particle. We observed that between the two HFF formulations, the smaller NPs (HFF2 = 71 nm) exhibited narrower distribution of intensity profile comparing to the larger NPs of HFF1 as shown in the Fig 3c (top and middle graph) and S3 Fig. Interestingly, we measured lower total fluorescence intensity for the smaller NPs formulated with the HFF method, however this difference was not observed in the bulk analysis, where the total amount of the encapsulated dye was similar

irrespectively the particle size. This result of the TIRF image analysis can be possibly explained by the difference in particle size, thus its loading capacity per particle and emitted fluorescence. On the other hand, the bulk analysis, which did not indicate this difference among the two particle sizes, can be explained by higher the total number of smaller NPs over the larger ones (per batch). That in summary gave similar fluorophore EE values when looking at the bulk.

The TIRF-based quantification of the similar diameter NPs formulated with the two different methods (microfluidic HFF1 and manual bulk) has shown alike distributions of collected fluorescence, with only slightly narrower profile for the HFF formulation. Similarly, to the previously observed heterogeneity in the size analysis, we observed uneven dye encapsulation among the particles. Again, in drug delivery, the homogenous drug distribution across all the particles imposes its controllability and better predictability *in vivo*, therefore the proposed characterization methods are crucial to assess these parameters.

### NPs incubation with MCF-7 cells

We evaluated our NPs formulated using the microfluidic chip with human breast adenocarcinoma epithelial cell line MCF-7, to confirm they remained non-toxic and retained favorable properties. Here we measured cell viability after 72h of exposure to the following: free DOX, NPs without cargo (placebo), NPs with encapsulated DiI and NPs loaded with DOX. In the Fig 4a we can see that the placebo and DiI loaded NPs (60 μg of polymer per well) did not affect the cell viability in comparison to the untreated cells (negative control). It indicates that the microfluidic formulation is a suitable method to obtain the NPs for drug delivery. On the other hand, unencapsulated Doxorubicin (80 ng of drug per well), which is known to intercalate into the genetic material, inducing cell apoptosis, had a major impact on the cell viability, with the result close to the positive control (cells exposed to Triton-X) [79]. The NPs loaded with Doxorubicin (DOX in the NPs: 8 ng of drug per well) induced some cell toxicity, indicating successful delivery of the cargo, however not as significant as the free drug. One of the reasons can be the lower Doxorubicin concentration per well, originating from the low EE for this drug (in the range of 10–15%), what resulted in the cells to be exposed to 10-times less compound. Secondly, the entrapped molecules are slowly released from the nanocarrier matrix, therefore the cell exposure to the drug is gradual, as the PLGA degrades over time [80, 81]. Overall, we observed that the NPs formulated with the microfluidic chip presented similar behavior in cell assay to the other PLGA-based NPs reported in the literature [25, 28].

Furthermore, we imaged the MCF-7 cells exposed to the microfluidic formulated NPs as can be seen in the Fig 4b and 4c). The cells were imaged after 24h of exposure to: free DiI, DiI-loaded NPs and placebo NPs, from left to right respectively. The free DiI stained whole cell membrane, meanwhile the dye loaded NPs were taken up by the cells and localized in distinctive spots, likely endosomes, as can be compared between the corresponding transmission and fluorescence images in Fig 4b and 4c. Similar observations were made for dye-loaded NPs, irrespectively of the NPs formulation technique (S4 Fig). In contrary to free DiI, the cells exposed to free DOX demonstrated the fluorescence concentrated mostly in the nuclei, and the DOX loaded NPs were localized outside (around nuclei), however due to the low EE of Doxorubicin in the NPs, we could not obtain satisfactory fluorescent images (S5 and S6 Figs). Future work could include follow up on the release studies of the encapsulated molecules, as well as incorporation of techniques enhancing DOX encapsulation, as mentioned above.

## Conclusions

We formulated several PLGA-PEG nanoparticles batches by manual bulk mixing and hydrodynamic flow focusing self-assembly at corresponding volumes of solvent and antisolvent,

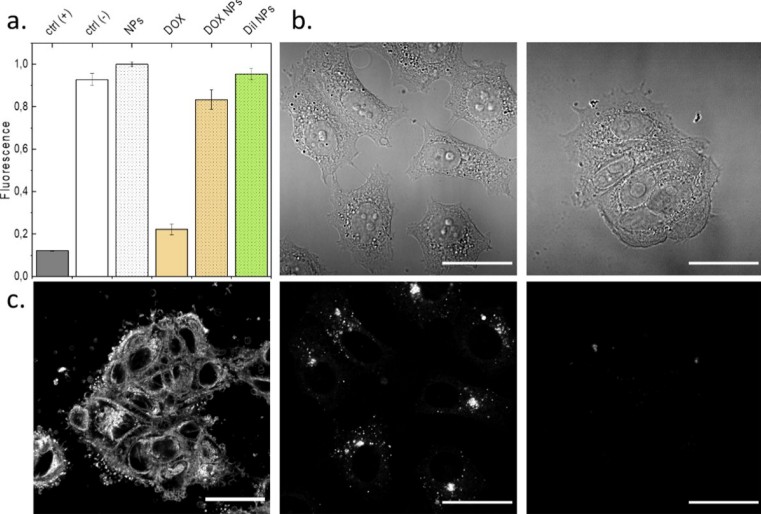

**Fig 4. Bioevaluation of formulated NPs.** a. Presto Blue Cell viability assay graph demonstrating viability of MCF-7 cells exposed to different NPs for 72h b. Confocal microscopy transmission images of MCF-7 cells incubated with DiI loaded NPs (left), and MCF-7 cells exposed to NPs without loading (right). c. Fluorescent confocal microscope images of the MCF-7 cells exposed to (left to right): DiI, DiI loaded NPs (corresponding the transmission image above), NPs without cargo (corresponding to the transmission image above). All scale bars 30 μm.

demonstrating that the adjustment of the solvent and antisolvent flow rates in confined mixing area allowed us to obtain broader range of diameters than with the bulk nanoprecipitation. The demonstrated NPs size tunability with the use of microfluidic chip highlights how a rational design could be executed in controlling certain properties of nanoparticles.

The studied encapsulation efficiency of three fluorescent compounds appeared to be dependent on the molecule type, and not the nanoprecipitation method, with DiI (most hydrophobic compound tested) having the highest EE. We used the TEM and TIRF techniques, to analyze individual particles in terms of their size and encapsulation of fluorescent molecules. They revealed heterogeneities among the particles for both studied properties, which could not be visualized within the bulk analysis methods. The encapsulation of DiI, analyzed for single particles, appeared similar among both formulation methods yielding larger NPs and more homogenous for microfluidic formulated NPs with smaller diameter. Finally, we demonstrated performing *in vitro* tests on MCF-7 cell line, that the microfluidic formulation does not alter biocompatibility of PLGA-PEG NPs.

# Materials and methods

## Solvent and antisolvent phase preparation

PLGA (L:G 50:50, Mw. 25–35 kDa; PolySciTech, Akina) and PLGA-PEG (L:G 50:50, Mw. 30 kDa and 5 kDa; PolySciTech, Akina) were weighed in a ratio 15% to 85%, respectively and dissolved in a solvent compatible with the microfluidic chip body and miscible with water: Acetonitrile (Chem-Lab, HPLC grade, Sigma Aldrich) resulting in final PLGA concentration of 10 mg/mL. The dissolution took place in a capped glass vial, at RT during 2h with 10 second of vortex each 20 minutes. Next, the solvent phase was filtered with 0,22 μm PTFE filter (Perkin Elmer, GC-Instruments) and used for the formulation.

For the antisolvent phase, freshly withdrawn purified water (MilliQ, Millipore) was filtered with a 0.22 μm sterile filter (Merck Millipore, Millex GP) and used.

## Fabrication of the PDMS chip

The 200 μm wide microchannel layout was designed in AutoCAD software and fabricated in an acetate photomask using CAD/Art Services (outputcity.com). The mask was used to prepare a master mold in the lithography process, briefly a glass substrate (microscope glass slide 25x75 mm, Corning) was washed with purified water and soap, flushed with EtOH, dried, treated with oxygen plasma (Expanded Plasma Cleaner PDC-002, Harrick Scientific Corporation). A negative SU-8 2100 photoresist (MicroChem) was deposited on the glass slide surface using 2-step spin coating to obtain a 60 μm-thick layer (following the instructions attached to the SU-8 photoresist). The deposition was followed by a 2-step soft bake (5 min. at 65°C, 20 min. at 95°C). To perform the photoresist polymerization the mask was placed on the top of the deposited layer, introduced into the UV-photolithography mask aligner (MJB4, SUSS Microtec) and exposed to 15.3 mW/cm$^2$ UV-light for 16 seconds. After UV irradiation the sample was post-baked for 5 minutes at 65°C and 10 minutes at 95°C. The unpolymerized photoresist was washed away with SU-8 developer (MicroChem) and the master mold dried and silanized with trichlorosilane (Sigma-Aldrich). The appearance of microstructures was examined using interferometer (Veeco Instruments, Wyko NT1100). Next, the PDMS replica was prepared by soft lithography molding: elastomer and curing agent (Sylgard 184, Dow Corning) were weighed in a plastic cup (in a ratio of 10:1, wt:wt) thoroughly mixed, degassed and poured over the master mold placed in a Petri dish. The dish with PDMS was left for 24h at RT and then placed into an oven at 60°C for 3h and then the cured PDMS replica was cut out, 0.8 mm holes punched out in the inlets/outlet, and bonded to a clean glass slide (25x75 mm, Corning).

## Bulk formulation

3 mL of the AS phase were added to a glass vial (5 mL) equipped with a magnetic stirring bar (VWR, size 8x3 mm) and placed on a magnetic stirring plate (IKA3050009 Big Squid White) at 100–200 rpm. Next the calculated volume of the S phase was added dropwise with a pipette (VWR 20–200 μL) into a stirring AS to obtain the desired S/AS. The nanoparticles suspension was let stirring for the next 5h to enhance the solvent evaporation. Afterwards the capped vial was stored in the fridge at 4–8°C until further analysis.

## Hydrodynamic flow focusing (HFF) formulation

PDMS microfluidic chip was connected to 3 syringes (one filled with S phase and two filled with the AS phase) using PTFE tubing (OD: 1,6 mm; ID 0,8 mm; Sigma Aldrich). The AS syringes (BD Plastic, 10 mL) were placed in a double syringe pump (New Era, NE-300), the S syringe (BD Plastic 2mL) was placed in a separate syringe pump (Chemyx Fusion 200). The chip was equipped with an outlet capillary directed into a collecting vial with a stirring bar for solvent extraction (as in the bulk method). The solvent flow rate was set at 5 μL/min or 15 μL/min and the AS flow rate was adjusted between 20 μL/min and 330 μL/min to screen the S/AS ratios in the range of 0.0075–0.11.

## Encapsulation of fluorescent compounds

1,1'-Dioctadecyl-3,3,3',3'-Tetramethylindocarbocyanine Perchlorate (DiI, lipophilic cationic carbocyanine dye, Sigma Aldrich, 42364) was weighed and dissolved in ACN, resulting in stock solution of 1.1 mM. Calculated amount of the stock was added to the S phase to reach the concentration of 7.1 μM.

Nile Red (Sigma Aldrich, 72485) was weighed and dissolved in ethanol, resulting in 0.46 mM stock solution, which was added to the solvent phase, reaching the final concentration of 7.1 μM. Doxorubicin Hydrochloride (Xing Chem ChemPharm) was weighed and dissolved in DMSO, resulting 1.04 mM stock solution, which was added to the solvent phase, reaching the final concentration of 7.1 μM. The nanoprecipitation in bulk or with the HFF chip was performed as described before.

## Nanoparticles concentration

Nanoparticles were washed and concentrated using ultrafiltration centrifugal filters (Amicon Ultra–0.5 mL. Ultracel, RC) with a nominal molecular weight limit 100 kDa. Briefly, the nanoparticles were filtered with 0.45 μm sterile filter (Merck Millipore, Millex HV 0.45 μm) and 400 uL of the suspension was directed into the purified water-rinsed Amicon filter. The filters were spun in a centrifuge (Eppendorf 5415 R) with the following parameters: 14kG, 5 min, 20˚C. After the centrifugation, the supernatant collected in the tube was removed and another 400 μL of NPs suspension added into the filter and the procedure repeated 3 times. Afterwards, the NPs were washed, resuspended in 80 μL of purified water and collected by placing the tube upside down in the microcentrifuge and spinning at 1 kG during 2 min.

## Encapsulation efficiency

DiI, Nile Red and Doxorubicin absorbance spectrum in acetonitrile was acquired using spectrophotometer (Infinite PRO M200, TECAN) and the maximum absorbance was found at: 550 nm, 538 nm and 480 nm respectively.

Fluorophore solutions for the calibration curve were prepared by dissolving the PLGA-PEG copolymer in ACN (10 mg/mL) and adding the corresponding fluorophore. The loaded NPs were concentrated as described previously, dissolved in acetonitrile and the absorbance was measured at previously determined wavelengths in a quartz cuvette (High Precision Cell Quartz SUPRASIL, Hellma Analytics, 10 mm).

The encapsulation efficiency was calculated as:

$$[EE]\% = \frac{Measured\ concentration}{Theoretical\ concentration} * 100\%$$

## NPs characterization—Dynamic light scattering

DLS (Malvern Zetasizer Nano—ZS) equipped with 633 nm laser and 173˚ detection optics, was used to measure the NPs size distribution in a colloidal suspension. The following SOP settings were used: Refractive index (RI): 1.460, Absorption: 0.0, Dispersant: water, viscosity: 0.887 cP, RI: 1.33, Temperature: 25˚C, equilibration time: 30 seconds, Cell: Quartz cuvettes ZEN2112. 50 μL of NPs suspension was added into the cuvette and the size measured in triplicate. Three independent batches were measured for each condition and the mean particle size value with standard deviation are reported. The cuvette was flushed 3x with purified water before each sample.

## NPs characterization—Transmission electron microscopy

The stock NPs were 4-fold diluted in purified water and then 10 μL of the NPs suspension was deposited on a carbon-coated copper grid (CF200-CU, 200 mesh, Electron Microscopy Sciences), washed 3x with 20 μL of purified water and stained with 10 μL of 2% uranyl acetate water solution (UB SCT). After the staining the excess liquid was blotted with a filter paper

and the grid placed into a desiccator for not less than 10h prior the image acquisition. The samples were imaged with a JEOL 1010 (Gatan, Japan) microscope equipped with a tungsten cathode (Electron Cryomicroscopy Unit from CCiTUB). The images were acquired at various magnifications (x30k–x120k) at 80kV with a CCD Megaview 1kx1k. The NPs diameter (of minimum 200 NPs/batch) was measured using ImageJ software.

## Comsol Multiphysics simulation

Comsol Multiphysics 5.3 software was used for the CFD simulations. The chip mesh was generated from the AutoCAD design (the same as used for the acetate mask). The simulation was performed for fluid laminar flow in stationary conditions to observe the transport of diluted species upon different flow rates. Water and Acetonitrile properties at RT and 1 atm were introduced to perform the computation. The parameters used for the computation were as following: dynamic viscosity of ACN: 0.389 mPas, the density of ACN: 0.786 g/mL, the properties of water were selected automatically from the software library.

## Total internal reflection (TIR) fluorescence imaging

30 μL of 10-fold diluted particles suspension was introduced into a flow chamber created with 24x24 mm glass cover slip (RS, France) attached with a two face Scotch tape to the edges of a glass slide (25x75 mm, Corning). The sample was incubated for 15 min. at RT and next 100 μL of purified water were introduced into the flow chamber to flush away unattached NPs.

Images were acquired using a Nikon N-STORM 4.0 system conFigd for total internal reflection (TIR) fluorescence, using a Perfect Focus System imaging. Excitation under the TIR conditions allowed to avoid illumination of out of focus, improving signal to noise ratio. DiI fluorophore was excited by illuminating the sample with a 5% power of 561 nm (80 mW) laser built into the microscope. During acquisition the integration time was 300 ms. Fluorescence was collected by means of a Nikon x100, 1.4 NA oil immersion objective and passed through a quad-band-pass dichroic filter (97335 Nikon). Images were recorded onto a 256 x 256-pixel region (pixel size 160 nm) of a Hamamatsu ORCA Flash 4.0 CMOS camera. The images were analyzed using ImageJ software. Briefly, the intensity threshold was set to filter the NPs in each image, and next the fluorescence intensity per particle was measured (for minimum 800 NPs) and plotted in a histogram graph.

## Cytotoxicity assay

Cell viability test was performed in triplicate, using PrestoBlue Cell Viability Kit (Invitrogen A13262) on MCF-7 cell line (ATCC) exposed to microfluidic device formulated NPs. The cells (p. 9) were cultured in a t-25 NUNC cell culture flask with Dulbecco's Modified Eagle Medium (DMEM, as received with L-Glutamine, 4.5 g/L D-glucose and pyruvate, Gibco) supplemented with FBS 10% (Gibco) and penicillin/streptomycin 1% (Biowest) at 37˚C and 5%CO2, until 70–80% confluency. Next, they were harvested using Trypsin-EDTA 0.25% (Gibco) and seeded in a 96-well plate (Nunclon Delta Surface, Thermo Scientific) at a density of 6k cells/well and incubated at 37˚C and 5% CO2. After 24 hours of incubation the cells were exposed for 72h to free Doxorubicin (~80 ng/well) and the following batches of the NPs (60 μg /well): without a cargo (placebo), with encapsulated DiI (~ 3ng/well) and with encapsulated Doxorubicin (~ 8 ng/well). All the NPs were formulated with the microfluidic chip at the same parameters. Untreated cells (in cell culture medium) were used as a negative control and cells with addition of 5% Triton-x were the positive control. In this work, cells viability was assessed by measuring the fluorescence value at the emission peak of resorufin. After the 72h the cells were washed with 1x PBS (Gibco) and the wells refilled with 100 μL of the cell culture media and

10 μL of PrestoBlue and further incubated for 1h 40 min. The fluorescence was measured at 590 nm using multimode plate reader (Infinite PRO M200, TECAN).

## Confocal microscopy

MCF-7 cells were seeded in a Lab-Tek (Nunc, Fisher Scientific) at density of 20k cells/well and incubated for 24h at 37˚C and 5% CO2. After the incubation to the wells were added: NPs without load (0,5 mg/well), NPs with DiI (polymer: 0,5 mg/well; dye: ~1.9 μg/well) and free DiI (2.5 μg/well) and further incubated for 20h. Next the cells were fixed with 4wt% solution of paraformaldehyde (PFA, Sigma Aldrich) in 1x PBS. After 10 minutes the fixative was washed away with 1x PBS. Cells were imaged using Confocal Microscope (Zeiss, LSM 800) with 63x oil immersion objective, pin hole 50 μm and pixel size of 50 nm. The fluorophore was excited with 561 nm laser at 0,20% with the emission detection in the range of 410–617 nm.

## Supporting information

**S1 File. Calculation of mixing time scale in the hydrodynamic flow focusing device.**
(DOCX)

**S1 Fig. Zeta potential measurement (Malvern, Zetasizer) for DiI loaded NPs and placebo NPs (no loading) for microfluidic and manual formulation.** The average values (given in the legend) are between -36.8 mV to -30.5 mV what agrees with reported values for this polymeric compound. The graphs and mean values represent an average of triplicates.
(TIF)

**S2 Fig.** TEM image of NPs with distinctive core and corona, a. formulated with HFF @ S/AS 0.036, b. formulated by manual bulk mixing @ S/AS 0.1.
(TIF)

**S3 Fig.** TIRF images of DiI loaded NPs formulated with HFF method, to compare fluorescence intensity between two formulations a. S/AS 0.095, particle diameter by DLS: 71m, b. S/AS 0.015, particle diameter by DLS 95 nm. Scale bar 5 μm.
(TIF)

**S4 Fig. Confocal microscopy images of MCF-7 cells incubated with DiI loaded NPs for 24h, NPs formulated by manual and microfluidic method.** Top row: fluorescence images and bottom row: corresponding transmission images, scale bars 30 μm.
(TIF)

**S5 Fig. TIRF images of DOX loaded NPs formulated with HFF method.** The particle fluorescence intensity is rather low due to the low EE, comparing for example to the S3 Fig. Scale bar 5 μm.
(TIF)

**S6 Fig. Confocal microscopy images of MCF-7 cells incubated for 24h with DOX loaded NPs, free DOX and placebo NPs (left to right).** Fluorescence signal is detected in cell cytoplasm and around the nucleus for NPS loaded with DOX. On the other hand the free form of DOX is mostly concentrated in the nuclei, as can be seen in the central panel. PLGE-PEG NPs (right panel) show slight fluorescent signal in the excitation/emission corresponding to the DOX. Scale bar 100 μm.
(TIF)

## Acknowledgments

AGM thanks to D. Izquierdo for the experimental support in the microfabrication at the MicroFabSpace and Microscopy Characterization Facility, Unit 7 of ICTS "NANBIOSIS" from CIBER-BBN at IBEC. We would like to thank the Electron Cryomicroscopy Unit (Scientific and Technological Centers Científics from Universitat de Barcelona, CCiTUB) for their technical support in electron microscopy.

## Author Contributions

**Conceptualization:** Josep Samitier Marti, Silvia Pujals, Lorenzo Albertazzi.

**Data curation:** Adrianna Glinkowska Mares, Gaia Pacassoni.

**Formal analysis:** Adrianna Glinkowska Mares.

**Methodology:** Adrianna Glinkowska Mares, Gaia Pacassoni.

**Project administration:** Lorenzo Albertazzi.

**Supervision:** Lorenzo Albertazzi.

**Writing – original draft:** Adrianna Glinkowska Mares.

**Writing – review & editing:** Adrianna Glinkowska Mares, Josep Samitier Marti, Silvia Pujals, Lorenzo Albertazzi.

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
