## [Decision Letter · Decision Letter 0]

25 Mar 2021

PONE-D-21-05079

Formulation of tunable size PLGA-PEG nanoparticles for drug delivery using microfluidic technology

PLOS ONE

Dear Dr. Albertazzi,

Thank you for submitting your manuscript to PLOS ONE. After careful consideration, we feel that it has merit but does not fully meet PLOS ONE’s publication criteria as it currently stands. Therefore, we invite you to submit a revised version of the manuscript that addresses the points raised during the review process.

We look forward to receiving your revised manuscript.

Kind regards,

José das Neves

Academic Editor

PLOS ONE

Journal Requirements:

PLOS requires an ORCID iD for the corresponding author in Editorial Manager on papers submitted after December 6th, 2016. Please ensure that you have an ORCID iD and that it is validated in Editorial Manager. To do this, go to ‘Update my Information’ (in the upper left-hand corner of the main menu), and click on the Fetch/Validate link next to the ORCID field. This will take you to the ORCID site and allow you to create a new iD or authenticate a pre-existing iD in Editorial Manager. Please see the following video for instructions on linking an ORCID iD to your Editorial Manager account: https://www.youtube.com/watch?v=_xcclfuvtxQ

Reviewers' comments:

Reviewer's Responses to Questions

**Comments to the Author**

1. Is the manuscript technically sound, and do the data support the conclusions?

Reviewer #1: Yes

Reviewer #2: Partly

Reviewer #3: Partly

2. Has the statistical analysis been performed appropriately and rigorously? 

Reviewer #1: Yes

Reviewer #2: No

Reviewer #3: I Don't Know

3. Have the authors made all data underlying the findings in their manuscript fully available?

Reviewer #1: Yes

Reviewer #2: Yes

Reviewer #3: No

4. Is the manuscript presented in an intelligible fashion and written in standard English?

Reviewer #1: Yes

Reviewer #2: Yes

Reviewer #3: Yes

5. Review Comments to the Author

Reviewer #1: In this work, the authors described the preparation of PLGA-PEG nanoparticles by microfluidic technique. I appreciate the manuscript that focuses on the research close to a real application, not only for the very sophisticated things, however, cannot be repeated or large scale synthesis is rather doubtful. This is an example of the paper in which the guidelines of well-known PLGA-PEG NPs preparation is described, however, in the microfluidic channels. Nevertheless, I have found some drawbacks that still required corrections. Therefore, after a major correction and required corrections, the manuscript could be published in PLOS ONE.

The author should discuss and compare their founding’s with these two papers already existing in the literature:

Journal of Colloid and Interface Science 475 (2016) 136–148

International Journal of Pharmaceutics 548 (2018) 530–539

Applied Nanoscience 8 (2018) 905–914

Why authors choose the acetonitrile (ACN) for the preparation of NPs? I recommend comparing the mixing parameters of ACN with water and other solvents as it was shown in the paper: Colloids and Surf. B 2021, 201, 111598.

Moreover, in future work, it is good to consider the more volatile solvent which is easier to remove from NPs, for instance, THF.

DiI, this abbreviation is not described in the section about the encapsulation of fluorescent compounds.

You should describe why there is so low DOX encapsulation?

I would recommend reading these two papers: Colloids Surfaces B Biointerfaces. 141 (2016) 187–195; Sci. Rep. 7 (2017) 1–12. I believe you can find the answer there.

I won’t start the sentence with: Thanks to…

I am confused with the results of cellular uptake. Why authors didn’t show the uptake for NPs loaded with DOX? Maybe, the reason for their low efficiency is correlated with the fact that these NPs cannot reach the nucleus of the cell?

Therefore, I will recommend repeating the preparation of NPs loaded with DOX to obtain the higher EE and, subsequently, their cellular uptake should be also conducted.

The relevant reviews in the field should be cited:

Adv Drug Deliv Rev. 2018 Mar 15;128:101-114, doi: 10.1016/j.addr.2017.12.015.

RSC Adv., 2019, 9, 2055-2072, DOI: 10.1039/C8RA08972H

Polym. Int. 68 (2019) 997-1014, https://doi.org/10.1002/pi.5753

Lab Chip, 2017, 17, 209-226, DOI: 10.1039/C6LC01049K

Nanoscale, 2016,8, 12430-12443, https://doi.org/10.1039/C5NR07964K

Reviewer #2: The synthesis of PLGA nanoparticles using microfluidics was compared to a bulk method of nanoparticle production. This has been done before for PLGA nanoparticles and so the authors should note the point of difference with their study and what has been done previously. A reference that is missing from this article that must be included is;

Streck S et al. (2019). Comparison of bulk and microfluidics for the formulation of functionalized poly-lactic-co-glycolic acid PLGA nanoparticles modified with cell-penetrating peptides of different architectures. International Journal of Pharmaceutics X https://doi.org/10.1016/j.ijpx.2019.100030

The authors encapsulated three different dye molecules and progressed to evaluate the toxicity of their formulation nanoparticles on a human breast epithelial cell line. This data was not mentioned in the abstract or conclusion however, and should be included.

How many independent batches were measured to obtain values for size and PDI of the nanoformulations?

Why was zeta-potential not measured?

The first paragraph of the introduction should be reduced because this information is VERY well established in the literature.

Specific comments

Abstract

Line 4. Review the use of English here “…allows to obtain…”.

Fig. 1. S and AS should be defined in the figure caption.

Fig. 2. HFF should be defined in the figure caption.

Paragraph “Particle diameter tunability using microfluidic chip” second sentence (“Faster diffusion yields smaller NPs…”). Please provide a reference citation or a citation to data in the manuscript for this statement.

“It is especially important to control the particle size, as their performance and cellular uptake depends on this feature.” Please provide a reference citation for this statement.

“This is a common strategy to encapsulate drugs into PLGA formulations.” Please provide reference citations for this statement.

Materials and methods, page 1, line 7 and elsewhere. Please correct ‘miliQ water’ to ‘purified water’.

Reviewer #3: The authors described the comparison of the PLGA-based nanoparticles prepared using the typical manual bulk mixing and a microfluidic chip-assisted nanoprecipitation. Peglated PLGA nanoparticles prepared by both methods have been widely documented in the literature. The authors claimed the tuneability of the particle size by microfluidic method. Again this is already very well-documented in the literature, particularly for PLGA nanoparticles. Therefore, it is difficult to extract the novelty of this study. It maybe beneficial for the authors to re-consider the focus of the study. Specific comments are shown below:

The first section of the Results and discussion, namely “Microfluidic hydrodynamic flow focusing and manual bulk formulation”, does not seem to be based on any data. It is not clear what the purpose of this section.

In this section, the authors claimed that CFD simulation was used to generate the Figure 1c. However, there was no CFD described in the method section. No real data to confirm the CFD prediction/simulation.

In section named “Particle diameter tunability using microfluidic chip” the authors only used one S/AS ratio for the conventional bulk method, which achieved similar particle size as the microfluidic method, but no other S/AS was used in the bulk method. This limited the quality of the comparison considering the significant differences in throughput rate of the two method. Such limited comparison seem inadequate for drawing any conclusion.

The cellular studies were only performed on microfluidic particles, but not particles prepared by bulk method. The EE and single particle analysis also indicated that there is no significant difference in bulk and microfluidic method when the same A/SA ratios were used. These findings that are directly relevant to the comparison are not mentioned in the conclusion.

No statistical analysis was described in the method section.

6. PLOS authors have the option to publish the peer review history of their article (what does this mean?). If published, this will include your full peer review and any attached files.

Reviewer #1: **Yes: **Marek Brzeziński

Reviewer #2: No

Reviewer #3: No

---

## [Author Response · Author response to Decision Letter 0]

21 Apr 2021

Response to reviewer was uploaded as a file (Cover letter + rebuttal)

---

## [Decision Letter · Decision Letter 1]

4 May 2021

Formulation of tunable size PLGA-PEG nanoparticles for drug delivery using microfluidic technology

PONE-D-21-05079R1

Dear Dr. Albertazzi,

We’re pleased to inform you that your manuscript has been judged scientifically suitable for publication and will be formally accepted for publication once it meets all outstanding technical requirements.

Kind regards,

José das Neves

Academic Editor

PLOS ONE

Additional Editor Comments (optional):

Reviewers' comments:

Reviewer's Responses to Questions

**Comments to the Author**

1. If the authors have adequately addressed your comments raised in a previous round of review and you feel that this manuscript is now acceptable for publication, you may indicate that here to bypass the “Comments to the Author” section, enter your conflict of interest statement in the “Confidential to Editor” section, and submit your "Accept" recommendation.

Reviewer #1: All comments have been addressed

2. Is the manuscript technically sound, and do the data support the conclusions?

Reviewer #1: Yes

3. Has the statistical analysis been performed appropriately and rigorously? 

Reviewer #1: Yes

4. Have the authors made all data underlying the findings in their manuscript fully available?

Reviewer #1: Yes

5. Is the manuscript presented in an intelligible fashion and written in standard English?

Reviewer #1: Yes

6. Review Comments to the Author

Reviewer #1: The authors made all required corrections. Therefore, I recommend the publication of this manusctipt.

7. PLOS authors have the option to publish the peer review history of their article (what does this mean?). If published, this will include your full peer review and any attached files.

Reviewer #1: **Yes: **Marek Brzeziński

---

## [Editor Report · Acceptance letter]

10 Jun 2021

PONE-D-21-05079R1 

Formulation of tunable size PLGA-PEG nanoparticles for drug delivery using microfluidic technology 

Dear Dr. Albertazzi:

I'm pleased to inform you that your manuscript has been deemed suitable for publication in PLOS ONE. Congratulations! Your manuscript is now with our production department. 

Kind regards, 

on behalf of

Dr. José das Neves 

Academic Editor

PLOS ONE